# Bootstrapped Transformer for Offline Reinforcement Learning

**Kerong Wang**[*]
Shanghai Jiao Tong University

**Hanye Zhao**
Shanghai Jiao Tong University

**Xufang Luo**
Microsoft Research Asia

**Kan Ren**[†]
Microsoft Research Asia

**Weinan Zhang**
Shanghai Jiao Tong University

**Dongsheng Li**
Microsoft Research Asia

## Abstract

Offline reinforcement learning (RL) aims at learning policies from previously collected static trajectory data without interacting with the real environment. Recent works provide a novel perspective by viewing offline RL as a generic sequence generation problem, adopting sequence models such as Transformer architecture to model distributions over trajectories, and repurposing beam search as a planning algorithm. However, the training datasets utilized in general offline RL tasks are quite limited and often suffer from insufficient distribution coverage, which could be harmful to training sequence generation In this paper, we propose a novel algorithm named Bootstrapped Transformer, which incorporates the idea of bootstrapping and leverages the learned model to self-generate more offline data to further boost the sequence model training. We conduct extensive experiments on two offline RL benchmarks and demonstrate that our model can largely remedy the existing offline RL training limitations and beat other strong baseline methods. We also analyze the generated pseudo data and the revealed characteristics may shed some light on offline RL training. The codes and supplementary materials are available at `https://seqml.github.io/bootorl`.

## 1  Introduction

Online reinforcement learning (RL) [42, 33, 8, 50] requires environments and interactions to train policies. However, such a requirement obstructs RL from being applied to many real-world problems where simulation environments are unavailable or interactions are incredibly costly. Offline RL tries to get rid of the requirement and learn policies from a previously collected offline dataset without interacting with the real environment. This setting enables more practical RL training in the real world and thus attracts lots of attention [27].

Unlike conventional approaches that focus on learning value functions [41] or computing policy gradients [34], recent works [32, 3] view the offline RL problem from a novel perspective, where it is considered a generic sequence generation problem. Specifically, these works utilize a mainstream sequence modeling neural network, i.e., Transformer [47], to model the joint distribution of the offline decision-making trajectory data, after discretizing each term vector within the transitions (states, rewards, actions), in a supervised learning (SL) manner. In such a sequence modeling framework, a trajectory consisting of states, actions, and rewards is viewed as a stream of tokens, and the learned Transformer can generate tokens in an autoregressive way and subsequently be utilized for execution in the real environments. This framework unifies multiple components in offline RL in one big

---

[*]The work was conducted during the internship of Kerong Wang and Hanye Zhao at Microsoft Research.
[†]Correspondence to Kan Ren, contact: kan.ren@microsoft.com.

sequence model, such as estimation of behavior policy and modeling for predictive dynamics [32], thus achieving superior performance.

However, there exist two major data limitations in offline RL tasks, which might be harmful to training sequence generation models yet have not been well addressed in the previous works. First, the offline data coverage is limited. The training data in offline RL tasks could be collected by arbitrary policies like human experts or random policies, which cannot guarantee sufficient coverage of the whole state and action spaces [11, 36]. The second issue lies in the amount of training data. Specifically, the largest dataset in the widely-used offline RL benchmark D4RL [9] contains only about 4,000 trajectories, which is quite limited especially considering those in natural language sequence modeling such as WikiText [31] which contains about 3,800,000 sentence sequences. Such limited datasets may deteriorate the sequence generation model, since the common belief in the literature is that training sequence generation models from scratch often requires a large amount of data with rich information [24]. Therefore, handling limited datasets should be carefully considered in the sequence modeling methods in offline RL tasks.

Some previous works try to tackle the above data limitations via data augmentation to expand the data coverage, either through image augmentation [25, 52, 51] such as cropping, or by applying perturbations on the state inputs [26, 43]. Nevertheless, these works can neither be applicable for other types of observations except image-based input, nor ensure to be consistent with the underlying Markov decision process (MDP). Another way is to train an additional model learning environment dynamics [18, 20, 30, 53]. Yet it requires extra efforts, and the goals of model learning and policy optimization have not been aligned, which may derive suboptimal solutions [54]. Moreover, all these methods have not considered the advanced sequence modeling framework.

In this paper, we try to relieve these problems under the sequence modeling paradigm for offline RL. Specifically, we consider *using the learned model itself to generate data and remedy the limitations in training datasets*. We further propose a novel algorithm, **Bootstrapped Transformer (BooT)**, to implement the above idea. Generally, BooT uses Transformer to generate data and then uses the generated sequence data of high confidence to further train the model for bootstrapping. To make our study comprehensive, we investigate two generation schemes. The first one performs autoregressive generation, and the other uses the teacher-forcing strategy [48]. We also test two bootstrapping schemes. One trains the model further on the generated data and discards them once they have been used; the other uses the generated data repeatedly during the later training process.

In this way, the above issues in previous works can be addressed simultaneously. Specifically, the sequence generation model unifies predictive dynamics models and policies [32]; thus, we do not need to train extra environment models for generating data, and such a generation-based method can also be applied to proprioceptive observations (e.g., positions & velocities information). Moreover, since the sequence model learns the distribution of the dataset and models the sequence generation probability, the data generated by the learned model itself will be more consistent with the original offline data [29]. Thus, we expect higher consistency of the pseudo trajectory data generated by the model than those of adding random noises, which has been empirically verified in our experiment.

We further conduct comprehensive experiments on multiple offline RL benchmarks, and BooT achieves significant performance improvements. For Gym domain in D4RL benchmark, BooT even reaches a much higher average score than one of the baselines with a pre-trained 80 times larger model [39], which strongly demonstrates the superiority of BooT. We also provide an in-depth analysis of data generated by BooT with both qualitative and quantitative results. Visualization demonstrates that the generated data act as an expansion of the original training dataset. Besides, we quantitatively measure the difference between the original and generated trajectories, and empirically conclude that the generated pseudo trajectories from BooT generally lie in a neighborhood of the corresponding original trajectories while keeping consistent with the underlying MDP in RL tasks, providing strong support for our motivation.

## 2 Related Work

**Offline Reinforcement Learning.**    Offline RL algorithms learn policies from a static dataset without interacting with the real environment [27]. Deploying off-policy RL algorithms directly into the offline setting suffers from the distributional shift problem, which could cause a significant performance drop, as revealed in [11]. To address this issue, model-free algorithms try to constrain the policy

closed to the behavior policy [11, 10] or penalize the values of out-of-distribution state-action pairs [23, 21]. On the other side, model-based algorithms [53, 20] simulate the real environment and generate more data for policy training. All these methods tend to require the policy to be pessimistic, while our algorithm does not involve this additional constraint. Besides, our algorithm is built upon the sequence generation framework, which is different from the above methods.

**Sequence Modeling.** Sequence modeling is a long-developed and popular area, and various methods have been proposed in natural language processing [16, 44], speech recognition [6], and information system [40]. Recently, Transformer [47] has illustrated powerful model performance and has been widely used in sequence modeling and generation tasks [5]. Some recent papers adapt Transformer into RL by viewing RL as a sequence modeling problem. Decision Transformer (DT) [3] learns the distribution of trajectories, and only predicts actions by feeding target rewards and previous states, rewards, and actions. Trajectory Transformer (TT) [32] further brings out the capability of the sequence model by predicting states, actions, and rewards and employing beam search. We leverage the paradigm of TT and provide a generation-based data augmentation methodology to improve the data coverage and further improve the performance. Recent works also try to use pre-trained Transformers on other domains and finetune on offline RL [39] or other decision-making [17, 28] tasks. We focus on the data augmentation method, which is different and compatible with the pre-training techniques.

**Data Augmentation (DA).** DA is a common approach to promoting data richness and enhancing modeling efficacy in supervised learning scenarios. They adopt data augmentation either through pseudo labeling [49], input perturbation [4], or generative model [15] to produce additional data. Although DA has not been widely used in RL tasks, recent works have shown the promising performance of DA in RL methods. Some of them target the data inefficiency issue. Specifically, RAD [26] incorporates common perturbation-based approaches, while DrQ [51], DrQ-v2 [52], and DrAC [37] also introduce regularization terms with respect to the augmented data in actor and critic. Some other works [14, 7, 13] try to achieve high stability for online RL via data augmentation. For offline RL, S4RL [43] provides various raw-state-based DA methods and takes similar regularization to DrQ. Different from the previous works, we augment the offline data distribution through sequence generation by the learned Transformer model to remedy the limitations in training datasets for better facilitating offline policy learning in this paper.

# 3 Preliminary

## 3.1 Online and Offline Reinforcement Learning

Generally, RL models the sequential decision making problem as a Markov Decision Process $\mathcal{M} = (\mathcal{S}, \mathcal{A}, \mathcal{T}, r, \gamma)$, where $\mathcal{S}$ is the state space and $\mathcal{A}$ is the action space. Given state $\boldsymbol{s}, \boldsymbol{s}' \in \mathcal{S}$ and action $\boldsymbol{a} \in \mathcal{A}$, $\mathcal{T}(\boldsymbol{s}'|\boldsymbol{s}, \boldsymbol{a})\colon \mathcal{S} \times \mathcal{A} \times \mathcal{S} \to [0, 1]$ is the transition probability and $r(\boldsymbol{s}, \boldsymbol{a})\colon \mathcal{S} \times \mathcal{A} \to \mathbb{R}$ defines the reward function. $\gamma \in (0, 1]$ is the discount factor. The policy $\pi\colon \mathcal{S} \times \mathcal{A} \to [0, 1]$ takes action $\boldsymbol{a}$ at state $\boldsymbol{s}$ with probability $\pi(\boldsymbol{a}|\boldsymbol{s})$. At time step $t \in [1, T]$, the accumulative discounted reward in the future, named reward-to-go, is $R_t = \sum_{t'=t}^{T} \gamma^{t'-t} r_{t'}$. The goal of *online* RL is to find a policy $\pi$ that maximizes $J = \mathbb{E}_{\boldsymbol{a}_t \sim \pi(\cdot|\boldsymbol{s}_t), \boldsymbol{s}_{t+1} \sim \mathcal{T}(\cdot|\boldsymbol{s}_t, \boldsymbol{a}_t)}[\sum_{t=1}^{T} \gamma^{t-1} r_t(\boldsymbol{s}_t, \boldsymbol{a}_t)]$ by learning from the transitions $(\boldsymbol{s}, \boldsymbol{a}, r, \boldsymbol{s}')$ through interacting with the environment in an online manner [45].

Offline RL, instead of interacting with the real environment, makes use of a fixed dataset $\mathcal{D}$ collected by behavior policy $\pi_b$, to learn a policy $\pi$ that maximizes $J$ for subsequent execution in the real environment. Here, $\pi_b$ can either be a single policy or a mixture of policies and is inaccessible. We assume that the collected data is trajectory-wise, i.e., $\mathcal{D} = \{\boldsymbol{\tau}_i\}_{i=1}^{D}$, where $\boldsymbol{\tau} = \{(\boldsymbol{s}_i, \boldsymbol{a}_i, r_i, \boldsymbol{s}_i')\}_{i=1}^{T}$.

## 3.2 Offline Reinforcement Learning as Sequence Modeling

In this paper, we take the offline RL problem as a sequence modeling task, following the previous work [32]. We introduce the details of the sequence modeling paradigm in this subsection. After that, we illustrate our data augmentation and bootstrapping strategy to remedy the limitations in training datasets and further improve the learning performance in Sec. 4.

---

**Algorithm 1** Training Procedure of BooT

---

**Input**: The original training set $\mathcal{D}$
**Parameter**: Number of epochs $E$, bootstrap epoch threshold $k$, generation percentage $\eta\%$ and generation length $T'$.
**Note**: Either BooT-o or BooT-r is executed.

1: **for** $i = 1, \ldots, E$ **do**
2:     Initialize the set of generated trajectories $\mathcal{G} = \emptyset$
3:     **for** each batch of trajectories $\boldsymbol{\tau} \subset \mathcal{D}$ **do**
4:         Train the model with $\boldsymbol{\tau}$ as Eq. (4)
5:         **if** $i > k$ **then**  # (perform bootstrapping)
6:             Generate $K$ new trajectories $\mathcal{G}' = \{\boldsymbol{\tau}'\}_1^K$ from $\boldsymbol{\tau}$ as Eq. (5) and Eq. (6)
7:             Select $\lfloor \eta\% \cdot K \rfloor$ trajectories with the highest confidence calculated as Eq. (9):
                $\mathcal{G}' \leftarrow \mathrm{argmax}_{S \subseteq \mathcal{G}', |S| = \lfloor \eta\% \cdot K \rfloor} \left( \sum_{\boldsymbol{\tau}' \in S} c_{\tau'} \right)$
8:             (BooT-o) Train the model with $\boldsymbol{\tau}' \in \mathcal{G}'$ as Eq. (4)
9:             (BooT-r) $\mathcal{G} \leftarrow \mathcal{G} \cup \mathcal{G}'$
10:         **end if**
11:     **end for**
12:     (BooT-r) $\mathcal{D} \leftarrow \mathcal{D} \cup \mathcal{G}$
13: **end for**

---

First, we treat each input trajectory $\boldsymbol{\tau}$ as a sequence and add reward-to-go $R_t = \sum_{t'=t}^{T} \gamma^{t'-t} r_{t'}$ after reward $r_t$ as auxiliary information at each timestep $t$, which acts as future heuristics for further execution planning,

$$\boldsymbol{\tau}_{\mathrm{aux}} = (\boldsymbol{s}_1, \boldsymbol{a}_1, r_1, R_1, \ldots, \boldsymbol{s}_T, \boldsymbol{a}_T, r_T, R_T) \ . \tag{1}$$

Second, a discretization approach is adopted to transform the continuous states and actions into discrete token spaces, each of which contains $N$ and $M$ tokens, respectively. This turns $\boldsymbol{\tau}_{\mathrm{aux}}$ into a sequence of discrete tokens with a total length of $(N + M + 2)T$ as

$$\boldsymbol{\tau}_{\mathrm{dis}} = \left( \ldots, s_t^1, s_t^2, \ldots, s_t^N, a_t^1, a_t^2, \ldots, a_t^M, r_t, R_t, \ldots \right) \ . \tag{2}$$

Without causing confusion, we will keep using $\boldsymbol{\tau}$ to indicate discretized augmented trajectory $\boldsymbol{\tau}_{\mathrm{dis}}$ in the following.

Finally, we model $\boldsymbol{\tau}$ with the Trajectory Transformer (TT) [32] from the perspective of sequence modeling. TT regards each trajectory as a sentence and is trained with the standard teacher-forcing procedure. Let $\theta$ be the parameters of TT and $P_\theta$ be its induced conditional probabilities. Define

$$\log P_\theta(\boldsymbol{\tau}_t | \boldsymbol{\tau}_{<t}) = \sum_{i=1}^{N} \log P_\theta(s_t^i | \boldsymbol{s}_t^{<i}, \boldsymbol{\tau}_{<t}) + \sum_{j=1}^{M} \log P_\theta(a_t^j | \boldsymbol{a}_t^{<j}, \boldsymbol{s}_t, \boldsymbol{\tau}_{<t})$$
$$+ \log P_\theta(r_t | \boldsymbol{a}_t, \boldsymbol{s}_t, \boldsymbol{\tau}_{<t}) + \log P_\theta(R_t | r_t, \boldsymbol{a}_t, \boldsymbol{s}_t, \boldsymbol{\tau}_{<t}) \ , \tag{3}$$

the training objective is to maximize

$$\mathcal{L}(\boldsymbol{\tau}) = \sum_{t=1}^{T} \log P_\theta(\boldsymbol{\tau}_t | \boldsymbol{\tau}_{<t}), \tag{4}$$

where $\boldsymbol{\tau}_{<t}$ is the trajectory from timestep 1 to $(t-1)$, $\boldsymbol{s}_t^{<i}$ is dimension 1 to $(i-1)$ of state $\boldsymbol{s}_t$, and the similar for $\boldsymbol{a}_t^{<j}$.

As for model inference, we adopt beam search as the planning algorithm to maximize cumulative discounted reward plus reward-to-go estimates, similar to [32].

## 4  Bootstrapped Transformer

In this section, we describe our Bootstrapped Transformer (BooT) algorithm, utilizing self-generated trajectories as auxiliary data to further train the model, which is the general idea of bootstrapping. Bootstrapping for supervised learning, such as pseudo labeling [19] and self-training [49], has been

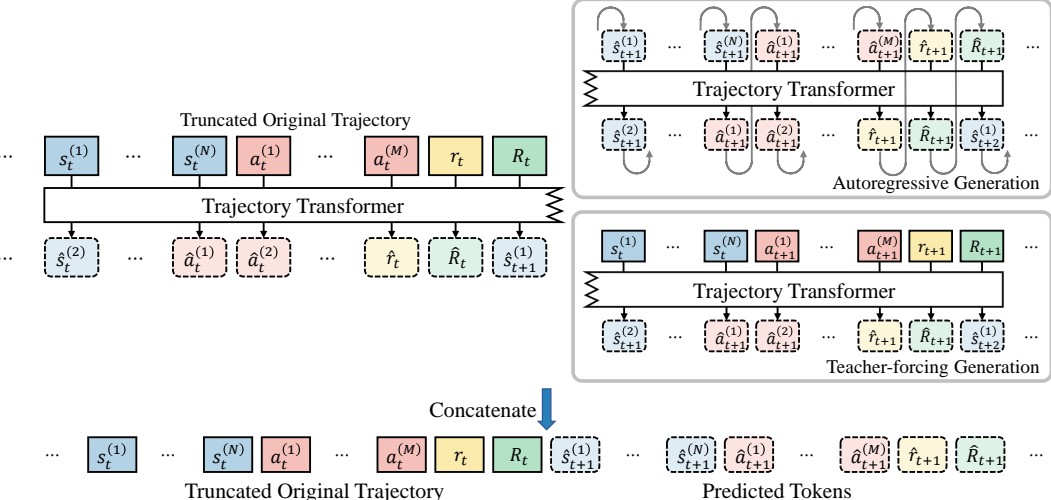

Figure 1: Illustration of trajectory generation. The generated trajectory is produced by concatenating the truncated original trajectory and the newly generated sequence by either of the two different generation schemes from the truncated position, as described in Sec. 4.1.

widely used. Apart from those using pseudo label generation, in the RL domain, we directly utilize the learned sequence model to generate novel trajectory data, which has been an expansion of the original offline data (as shown in Sec. 5.3), and we further augment the original data distribution using the generated data to better approximate the real distribution following the underlying MDP. The self-generated trajectories are fed as additional training data to further boost sequence modeling and decision making.

Overall, we divide BooT into two main parts, (i) generating trajectories with the learned sequence model and (ii) using the generated trajectories to augment the original offline dataset for bootstrapping the sequence model learning. The overall training algorithm of BooT has been described in Algorithm 1.

## 4.1 Trajectory Generation

In this section, we demonstrate the details of trajectory generation. For an original trajectory $\boldsymbol{\tau} \in \mathcal{D}$ in the original dataset, we resample the last $T'$ timesteps ($T' < T$) as

$$\tilde{\boldsymbol{\tau}}_{>T-T'} = \left( \tilde{\boldsymbol{s}}_{T-T'+1}, \tilde{\boldsymbol{a}}_{T-T'+1}, \ldots, \tilde{R}_T \right) \sim P_\theta \left( \boldsymbol{\tau}_{>T-T'} | \boldsymbol{\tau}_{\leq T-T'} \right) , \tag{5}$$

where $\boldsymbol{\tau}_{\leq T-T'}$ represents the original trajectory truncated at timestep $(T-T')$. Then, we concatenate the generated token sequence to the truncated original trajectory and obtain a new trajectory

$$\boldsymbol{\tau}' = \boldsymbol{\tau}_{\leq T-T'} \circ \tilde{\boldsymbol{\tau}}_{>T-T'} , \tag{6}$$

where $\circ$ denotes concatenation operation.

We investigate two different trajectory generation schemes below, both of which are demonstrated in Figure 1. We briefly compare these two methods in both efficiency and generation quality here, and perform detailed comparison experiments in Sec. 5.3.

1. **Autoregressive generation** is to produce each predicted token $\tilde{y}_n$ by sampling from the probability conditioned on the previously generated tokens as

$$\tilde{y}_n \sim P_\theta \left( y_n | \tilde{\boldsymbol{y}}_{<n}, \boldsymbol{\tau}_{\leq T-T'} \right) . \tag{7}$$

Here we denote the $n^{\text{th}}$ generated token as $y_n$ and the sequence of all the previously generated tokens as $\tilde{\boldsymbol{y}}_{<n}$. The autoregressive generation scheme requires recurrent predictions based on the previously generated tokens.

2. **Teacher-forcing generation** adopts a similar procedure as teacher-forcing training by conditioning on the original token ground truth from the previous timesteps

$$\tilde{y}_n \sim P_\theta \left( y_n | \boldsymbol{y}_{<n}, \boldsymbol{\tau}_{\leq T-T'} \right) . \tag{8}$$

Here, $\boldsymbol{y}_{<n}$ denotes tokens from the original trajectory with position previous to $\tilde{y}_n$. Teacher-forcing generation only relies on ground truth tokens.

Due to error accumulation in the sequential generation process, autoregressive generation tends to produce trajectories that deviate more from the original data distribution than teacher-forcing generation. As has been found in Sec. 5.3, autoregressive generation could expand the coverage of the training data more effectively and might derive better learning efficacy. Compared to autoregressive generation, teacher-forcing requires only one forward pass and thus consumes less time. However, since teacher-forcing generation is not able to expand dataset coverage as effectively as autoregressive generation, it might lead to slightly less improvement in performance.

### 4.2 Training on the Augmented Dataset

After generating new trajectories, we use these trajectories to train our sequence model in addition to the original trajectory data.

In BooT, we conduct two different bootstrapping schemes: Bootstrap-once, denoted as **BooT-o**, and Bootstrap-repeat, denoted as **BooT-r**. As shown in Algorithm 1, both methods will first train the sequence model over the original offline trajectories and then utilize it to generate novel trajectories. After that, BooT-o will train the model again on the generated trajectories and discard them *immediately* once they have been used. In contrast, BooT-r will append the generated trajectories $\mathcal{G}$ into the dataset $\mathcal{D}$, and the generated trajectories will be used in every epoch after they are appended to the dataset.

In order to prevent accumulating learning bias caused by training with inaccurately generated data, we choose a part of the trajectories with the *highest* confidence scores in each batch according to the generation percentage $\eta\%$. Here, we define confidence as the average log probability of all the generated tokens as

$$c(\boldsymbol{\tau}) = \frac{1}{T'(N+M+2)} \sum_{t=T-T'+1}^{T} \log P_\theta(\boldsymbol{\tau}_t | \boldsymbol{\tau}_{<t}) , \tag{9}$$

where $\log P_\theta(\boldsymbol{\tau}_t | \boldsymbol{\tau}_{<t})$ is defined in Eq. (3). Intuitively, since the confidence score of trajectory is defined by its prediction probability that models the sequence generation mechanism, by filtering the generated trajectories with their confidence score, we can keep only the generated trajectory data of a high quality to further help the sequence model training through feeding these quality data back for bootstrapping.

Moreover, to stabilize bootstrapped training, we leverage the idea of *curriculum learning* described in [2] but adopt a more straightforward method here. The sequence model is first trained over the original offline dataset and then gradually learns from the self-generated trajectories. To achieve this, we only use the generated trajectories to train our model after training $k$ epochs over the original trajectories (without generated data). Though we could use more sophisticated scheduled sampling strategies such as [2], they may incorporate additional hyperparameters, resulting in complex parameter tuning. It turns out that our simple proposed version has shown promising effectiveness in the experiments.

## 5 Experiments

In this section, we mainly (i) evaluate the performance of BooT and (ii) investigate the characteristics of the generated trajectories of BooT. We first compare the evaluation results of our algorithm to other offline RL algorithms, and conduct a comparison among different bootstrapping and sequence generation schemes. Then we study the characteristics of generated data with both qualitative and quantitative analysis to illustrate more insights of our method. Finally, we introduce the ablation study on hyperparameters and analyze their influence on BooT.

Table 1: Results on D4RL dataset. BooT performs best on 5 out of 9 datasets compared to other baselines and obtained the highest average score of 9 datasets. For results of TT (Reproduce, Re-train, +S4RL (All), +S4RL (Last)) and BooT, we report the mean and standard deviation corresponding to 15 random seeds (5 training seeds for independently trained Transformers and 3 evaluation seeds for each model).

| Dataset | Environment | BC | MBOP | CQL | DT Original | DT GPT2 | TT Original | TT Reproduce | TT Re-train | TT +S4RL (All) | TT +S4RL (Last) | BooT |
|---|---|---|---|---|---|---|---|---|---|---|---|---|
| Medium-Expert | HalfCheetah | 59.9 | **105.9** | 91.6 | 86.8 | 91.8 | 40.8 | $81.4 \pm 22.8$ | $90.5 \pm 12.4$ | $87.5 \pm 16.8$ | $84.3 \pm 20.3$ | $94.0 \pm 1.0$ |
| Medium-Expert | Hopper | 79.6 | 55.1 | 105.4 | 107.6 | **110.9** | 106.0 | $66.2 \pm 23.4$ | $67.0 \pm 22.7$ | $92.6 \pm 27.8$ | $96.9 \pm 23.1$ | $102.3 \pm 19.4$ |
| Medium-Expert | Walker2D | 36.6 | 70.2 | 108.8 | 108.1 | 91.0 | 108.9 | $102.2 \pm 12.8$ | $99.1 \pm 25.6$ | $107.6 \pm 7.4$ | $100.5 \pm 27.5$ | $\mathbf{110.4} \pm 2.0$ |
| Medium | HalfCheetah | 43.1 | 44.6 | 44.0 | 42.6 | 44.0 | 42.8 | $46.1 \pm 1.3$ | $44.1 \pm 8.8$ | $46.7 \pm 1.4$ | $46.0 \pm 1.2$ | $\mathbf{50.6} \pm 0.8$ |
| Medium | Hopper | 63.9 | 48.8 | 58.5 | 67.6 | 67.4 | **79.1** | $67.1 \pm 20.1$ | $64.9 \pm 19.2$ | $53.9 \pm 10.6$ | $56.0 \pm 14.7$ | $70.2 \pm 18.1$ |
| Medium | Walker2D | 77.3 | 41.0 | 72.5 | 74.0 | 81.3 | 78.3 | $71.1 \pm 25.7$ | $81.6 \pm 6.1$ | $80.2 \pm 8.6$ | $80.0 \pm 10.8$ | $\mathbf{82.9} \pm 11.7$ |
| Medium-Replay | HalfCheetah | 4.3 | 42.3 | 45.5 | 36.6 | 40.3 | 44.1 | $43.1 \pm 1.6$ | $36.5 \pm 15.0$ | $43.1 \pm 4.1$ | $40.2 \pm 12.7$ | $\mathbf{46.5} \pm 1.2$ |
| Medium-Replay | Hopper | 27.6 | 12.4 | 95.0 | 82.7 | 94.4 | **99.4** | $86.4 \pm 25.1$ | $90.8 \pm 24.0$ | $76.1 \pm 27.0$ | $72.8 \pm 29.9$ | $92.9 \pm 13.2$ |
| Medium-Replay | Walker2D | 36.9 | 9.7 | 77.2 | 66.6 | 72.7 | 79.4 | $66.9 \pm 35.5$ | $50.3 \pm 28.4$ | $52.1 \pm 40.7$ | $61.1 \pm 37.8$ | $\mathbf{87.6} \pm 9.9$ |
| **Average** | | 47.7 | 47.8 | 77.6 | 74.7 | 80.1 | 72.6 | 70.1 | 69.4 | 71.1 | 70.9 | **81.9** |

## 5.1 Evaluation Settings

**Benchmark and Compared Baselines.** We evaluate our BooT algorithm on the dataset of continuous control tasks from the D4RL offline dataset [9]. We first perform experiments on Gym domain, which includes three environments (halfcheetah, hopper, walker2d) with three levels (medium, medium-replay, medium-expert) for each environment. We compare BooT to the following various baselines: (1) behavior cloning (**BC**), a pure imitation learning method; (2) model-based offline planning (**MBOP** [1]), a current strong baseline with model-based offline RL algorithm; (3) conservative Q-learning (**CQL** [23]), a current strong baseline of model-free offline RL algorithm; (4) Decision Transformer (**DT Original** [3]) with its variant using GPT2 pre-trained model (**DT GPT2**, [39]), and (5) Trajectory Transformer (**TT** [32]), two recent works applying Transformer to offline RL; and multiple variants derived from our proposed method BooT for ablation studies. In addition, we perform experiments on Adroit domain [38] in D4RL dataset, including four environments (pen, hammer, door, relocate) with also three levels (human, cloned, expert) for each environment to better compare our method with TT and CQL baselines.

We report the normalized scores according to the protocol in [9], where a normalized score of 0 corresponds to the average return of random policies and 100 corresponds to that of a domain-specific expert. The performance of BC is taken from [3], and the performance of CQL is taken from [22]. The results of MBOP, DT Original, DT GPT2, and TT are reported from their original papers.

Besides, we reproduce the TT algorithm, namely *TT (Reproduce)*. We also perform an additional experiment with a slightly modified setting as *TT (Re-train)*: after the $k^{\text{th}}$ original training epoch, for each training batch, we randomly choose $\lfloor \eta\% \cdot K \rfloor$ original trajectories and train the model for the second time. This setting is to ensure the numbers of training steps are the same for TT and BooT.

To compare with other offline data augmentation methods, we perform two additional experiments using TT with S4RL [43] techniques, which adds random noise to the input trajectories as augmented data. Specifically, *TT+S4RL (All)* adds random noise to all states in a trajectory, while *TT+S4RL (Last)* adds random noise to the last $T'$ timesteps of a trajectory. The former setting resembles adding random noise to raw states in S4RL, while the latter is to compare with BooT where generated trajectories only differ from the original trajectories at the last $T'$ timestep.

For a fair comparison, we adopt the same training hyperparameters with TT in all the above experiments. We also use the same model structure and model size as TT for all BooT experiments.

Finally, we compare the performances of different bootstrapping schemes for BooT, using BooT-o and BooT-r, with autoregressive and teacher-forcing trajectories, respectively. All results in this section correspond to 15 random seeds, including 5 training seeds for independently trained Transformer models and 3 evaluation seeds for each model, if not explicitly mentioned.

## 5.2 Detailed Results

In this section, we first present the overall result of BooT compared with other baselines in Table 1, and then show the detailed results of BooT with different bootstrapping schemes in Table 2. As is shown in Table 1, BooT achieves improvements from TT and performs on par or better than other

Table 2: Comparison between different trajectory generation and bootstrap schemes on Gym domain. BooT-o and BooT-r at the header refer to bootstrapping schemes, and Autoregressive (AR) and Teacher-forcing (TF) refer to trajectory generation schemes.

| Dataset | Environment | BooT-o, AR | BooT-r, AR | BooT-o, TF | BooT-r, TF |
|---|---|---|---|---|---|
| Med-Expert | HalfCheetah | 94.0 $\pm$ 1.0 | 94.5 $\pm$ 0.9 | 93.6 $\pm$ 1.3 | 95.2 $\pm$ 1.0 |
| Med-Expert | Hopper | 102.3 $\pm$ 19.4 | 88.7 $\pm$ 29.8 | 96.1 $\pm$ 27.5 | 94.6 $\pm$ 24.5 |
| Med-Expert | Walker2d | 110.4 $\pm$ 2.0 | 111.1 $\pm$ 1.0 | 111.0 $\pm$ 1.8 | 111.3 $\pm$ 1.2 |
| Medium | HalfCheetah | 50.6 $\pm$ 0.8 | 51.4 $\pm$ 1.0 | 50.1 $\pm$ 1.3 | 51.3 $\pm$ 0.7 |
| Medium | Hopper | 70.2 $\pm$ 18.1 | 71.2 $\pm$ 20.4 | 71.1 $\pm$ 18.8 | 70.4 $\pm$ 17.1 |
| Medium | Walker2d | 82.9 $\pm$ 11.7 | 84.0 $\pm$ 8.3 | 85.7 $\pm$ 3.1 | 84.6 $\pm$ 3.6 |
| Med-Replay | HalfCheetah | 46.5 $\pm$ 1.2 | 46.4 $\pm$ 1.2 | 45.6 $\pm$ 2.1 | 46.6 $\pm$ 1.3 |
| Med-Replay | Hopper | 92.9 $\pm$ 13.2 | 91.3 $\pm$ 17.8 | 89.1 $\pm$ 26.3 | 96.8 $\pm$ 11.9 |
| Med-Replay | Walker2d | 87.6 $\pm$ 9.9 | 73.7 $\pm$ 25.9 | 77.1 $\pm$ 24.6 | 77.2 $\pm$ 24.1 |
| **Average** | | 81.9 | 79.1 | 79.9 | 80.9 |

offline RL algorithms. Compared to our reproduced results of TT, BooT obtains improvements by 16.8% in average score, and achieves better performance on all 9 datasets.

Comparison among TT (Reproduce), TT (Re-train), and BooT suggest the improvement in the performance of BooT does not result from the extra training steps of the algorithm. Furthermore, we observe that directly adding Gaussian noise to the training data does not help improve the performance effectively as BooT from the results of TT+S4RL experiments. We argue that data generated by the model itself are more consistent with the underlying MDP than by adding noise; thus, BooT obtains larger performance improvements, which also echos to our motivation.

Experiment results in Table 3 indicate that BooT can also help improve the

Table 3: Comparison between TT and CQL baselines and variants for BooT on Adroit domain. We list the mean and standard deviation corresponding to 15 random seeds. Note that results of CQL are taken from the original paper and they did not provide performances on expert datasets.

| Dataset | Environment | CQL | TT | BooT-o, TF | BooT-r, TF |
|---|---|---|---|---|---|
| Expert | Pen | - | 72.0 $\pm$ 62.9 | 72.3 $\pm$ 64.4 | 79.7 $\pm$ 60.0 |
| Expert | Hammer | - | 15.5 $\pm$ 39.7 | 0.7 $\pm$ 1.0 | 0.8 $\pm$ 1.0 |
| Expert | Door | - | 94.1 $\pm$ 28.5 | 93.3 $\pm$ 32.8 | 106.5 $\pm$ 3.4 |
| Expert | Relocate | - | 10.3 $\pm$ 18.6 | 5.3 $\pm$ 9.4 | 10.0 $\pm$ 13.9 |
| Human | Pen | 55.8 | 36.4 $\pm$ 66.0 | 28.4 $\pm$ 52.9 | 54.3 $\pm$ 62.8 |
| Human | Hammer | 2.1 | 0.8 $\pm$ 0.6 | 0.7 $\pm$ 0.6 | 0.7 $\pm$ 0.7 |
| Human | Door | 9.1 | 0.1 $\pm$ 0.1 | 0.1 $\pm$ 0.0 | 0.6 $\pm$ 1.6 |
| Human | Relocate | 0.4 | 0.0 $\pm$ 0.0 | 0.0 $\pm$ 0.1 | 0.0 $\pm$ 0.1 |
| Cloned | Pen | 40.3 | 11.4 $\pm$ 42.5 | 31.5 $\pm$ 68.1 | 31.0 $\pm$ 68.5 |
| Cloned | Hammer | 5.7 | 0.5 $\pm$ 0.5 | 0.3 $\pm$ 0.2 | 0.5 $\pm$ 0.6 |
| Cloned | Door | 3.5 | -0.1 $\pm$ 0.1 | -0.0 $\pm$ 0.1 | -0.0 $\pm$ 0.1 |
| Cloned | Relocate | -0.1 | -0.1 $\pm$ 0.1 | -0.0 $\pm$ 0.1 | -0.0 $\pm$ 0.1 |
| **Average** | | 14.6 | 20.1 | 19.4 | **23.7** |

performance on more complicated tasks in Adroit domain, where BooT achieves better performance than TT baseline with an average improvement of 17.9%. Results show that teacher-forcing generation has already obtained noticeable improvement compared to the baselines, and autoregressive generation consumes too much time in Adroit domain as the number of dimensions of the trajectories is too large. As a result, we only perform experiments with teacher-forcing generation on Adroit domain.

## 5.3 Further Analysis of Bootstrapping

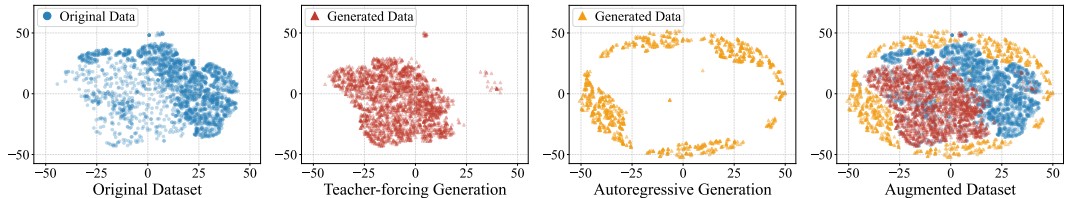

Figure 2: Distribution of $(s, a, r, R)$ transitions visualized with t-SNE in one example environment. Blue points correspond to $(s, a, r, R)$ transitions from the halfcheetah-medium dataset, red points correspond to generated data with teacher-forcing trajectory generation, and yellow points correspond to autoregressive trajectory generation. Generated data are drawn from the last training epoch. Best viewed in color.

To better understand how BooT helps improve the performance, we perform numerical analysis on the generated trajectories in Table 4. We calculate the distance from the generated trajectories of BooT to their corresponding original offline trajectories, denoted as *Dataset*, and the distance from the generated trajectories to real trajectories from the environment in the eval-

Table 4: Comparison of different distance metrics on generated trajectories and original trajectories. We list the mean value on 9 datasets.

| | Metric | TT + S4RL | BooT-o, AR | BooT-o, TF |
|---|---|---|---|---|
| | **Avg. performance** | 71.1 | 81.9 | 79.9 |
| *Dataset* | RMSE | 21.5 | 16.2 | 8.9 |
| | MMD ($\times 10^{-3}$) | 8.7 | 26.2 | 9.8 |
| *Environment* | RMSE | 20.0 | 14.8 | 14.7 |
| | MMD ($\times 10^{-3}$) | 33.1 | 7.6 | 12.2 |

uation phase, denoted as *Environment*, compared to those from TT+S4RL method. As we are investigating generation results but not training schemes here, we calculate the distance using only BooT-o algorithm, which is sufficient for analysis. *RMSE* stands for the *Root Mean Squared Error* value of the trajectory difference, and *MMD* stands for the *Maximum Mean Discrepancy* of two trajectory sets. More details can be found in Appendix D.1.

In Table 4, the results of *Dataset* show that BooT produces trajectories with smaller RMSE compared to adding random noise, which demonstrates that generated trajectories generally lie in a neighborhood of the corresponding original offline trajectories in Euclidean space. Also, trajectories generated by BooT have larger MMD compared with the baseline, showing that generated data are different from original offline trajectories w.r.t. the data distribution. This indicates that BooT expands the original offline data, and the visualization in Figure 2 also verifies this claim. But is such expansion reasonable? The answer is yes and is supported by *Environment* results, which show that the distance between the generated trajectories and real trajectories from the evaluation environment is relatively smaller compared with the baseline, demonstrating that the generated trajectories are more consistent with the real distribution of the underlying MDP in RL tasks. Thus, BooT provides a more reasonable expansion to the original offline data, and subsequently improves the performance of the model.

Furthermore, in Figure 2, we visualize the distribution of (i) transitions $(\boldsymbol{s}, \boldsymbol{a}, r, R)$ in the original data, (ii) data from teacher-forcing generation, and (iii) data from autoregressive generation in BooT. We excerpt the last $T'$ transitions from these trajectories, reduce them to 2-dimension via t-SNE [46] altogether, and plot them separately. We also provide more visualization in Appendix D.2.

It is clearly illustrated that a large portion of data generated by the teacher-forcing method overlaps with the support of the original dataset, while those generated autoregressively lie out of the original data distribution. The overall results demonstrate that generated trajectories from BooT expand the data coverage while still keeping consistency with the underlying MDP of the corresponding RL task, thus resulting in better performance compared to the baselines. This also suggests why BooT does not require extra mechanisms such as conservatism [1] to work well.

## 5.4 Ablation on Hyperparameters

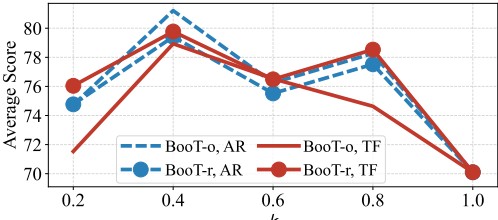

Figure 3: Ablation experiment results on bootstrap epoch threshold $k$ as shown in Algorithm 1.

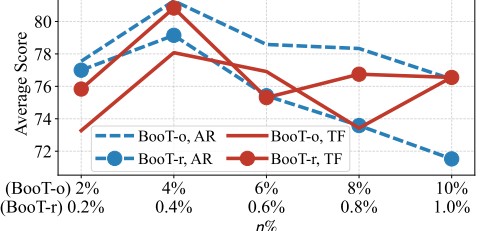

Figure 4: Ablation experiment results on generation percentage $\eta\%$ as shown in Algorithm 1.

**When to start to perform bootstrapping?** We performed an ablation study on generation threshold $k$, which determines the threshold to start performing bootstrapping. For $k = 0.2$, BooT will perform bootstrapping after the model is trained without generating sequences for $20\%$ of training epochs. As is shown in Figure 3, the performance of BooT will first increase and then decrease as $k$ increases. It is reasonable that if bootstrapping starts too early, the model will generate sequences that differ from the training dataset, and accumulate more error in the following training procedure. If bootstrapping starts too late, the model will learn little from bootstrapping as the model may have converged. Thus, an appropriate value of $k$ could be selected.

**How many new trajectories to generate in bootstrapping?** We conduct an ablation study on generation percentage $\eta\%$, which is used to control the ratio of the generated trajectories to the original offline trajectories. As shown in Figure 4, the performance of BooT first increases and then decreases as $\eta\%$ increases. For a small $\eta\%$ value, the number of total training steps on generated data might be too small to effectively improve the performance of the model. However, a large $\eta\%$ value may involve too much generated data with a low confidence score. As explained in Sec. 4.2, this could cause accumulating learning bias and results in a performance drop.

**How long should new trajectories be generated?** We perform an additional ablation study on generation length $T'$, which is used to control the length of generated sequences. As shown in Figure 5, the performance of BooT with TF generation generally decreases as $T'$ increases, and most results are better than the baselines. Furthermore, the performance of BooT-r is relatively robust to the changes of $T'$. As the number of tokens per timestep differs among different environments, it is difficult to determine the most suitable $T'$ for all environments. Besides, $T' = 1$ performs the best in experiments, so we think it is sufficient to simply set $T' = 1$ for most cases.

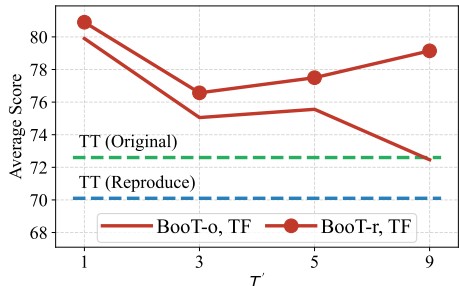

Figure 5: Ablation results on generation length $T'$ with TF generation, where original trajectory length $T = 10$.

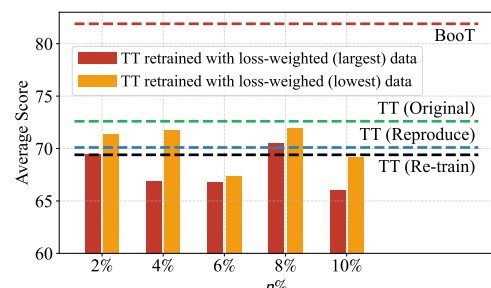

Figure 6: TT-retrain results with the lowest / largest $\eta\%$ loss in each batch. We report the mean results corresponding to 15 random seeds.

## 5.5 Experiments of TT Retrained with Loss-weighted Data

We implement an additional TT baseline with loss-weighted retraining, to experiment whether the generated data are better for bootstrapping. In the experiment, we add a simple loss-weighted re-train component for TT-retrain, choosing the data with the lowest / largest $\eta\%$ loss in each batch to re-train the model, to better compare with TT and TT-retrain baselines. The results are shown in Figure 6. For supplementary, the numerical results are provided in Appendix C.4.

The results have illustrated that using a loss-weighted re-train does help to improve the performance of the model, but BooT still performs much better than both schemes of loss-weighted re-train. Compared to simple random re-training scheme, re-training the model with data corresponding to the lowest loss helps improve the performance of the model, while using data corresponding to the largest loss has less improvement or even possibly degrades the performance.

## 6 Conclusion and Future Work

In this paper, we consider the data coverage issues encountered in offline RL scenarios and propose a bootstrapping algorithm with a sequence modeling paradigm to self-generate novel trajectory data and in turn feed them back to boost sequence model training. We adopt two sequence generation schemes, and each of them generates trajectories with unique characteristics compared to the original offline data distribution. Compared to the other strong baselines, the experiments over the public offline RL benchmarks have illustrated the great effectiveness of our proposed method, and the analysis also provides some insightful evidence about how it works.

One drawback of our method is the relatively larger training time consumption compared to other methods because of online pseudo training data generation. In the future, we plan to improve the efficiency by incorporating more advanced techniques such as non-autoregressive sequence generation [12, 35], to improve the training efficiency.

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
