# OpenReview forum: "Bootstrapped Transformer for Offline Reinforcement Learning"
_NeurIPS.cc/2022/Conference — NeurIPS 2022 Accept_

### Official Review · Reviewer_i5fj · 2022-07-10

**Rating:** 6
**Confidence:** 5
**Soundness:** 3 good
**Presentation:** 3 good
**Contribution:** 3 good

**Summary:**

This paper extends recent sequence-modeling approaches to RL with model-based data augmentation, with the end result looking like a Dyna-inspired [[Sutton 1991](https://dl.acm.org/doi/10.1145/122344.122377)] variant of the Trajectory Transformer. A few design decisions, such as the sampling strategy (autoregressive versus teacher-forced) and how often to reuse the model-generated data, are investigated. The algorithm is evaluated on the locomotion and adroit domains of the D4RL offline RL benchmark.

**Questions:**

1.  _"However, since teacher-forcing generation is not able to expand dataset coverage as effectively as autoregressive generation, it will lead to less improvement in performance."_

This explanation is what I would have expected, but does it actually pan out in the results? Table 2 shows teacher forcing (TF) underperforming autoregressive (AR) generation with “-once” sampling, but (TF) outperforming (AR) generation with “-repeat” sampling. Overall, though, it’s all a bit of a wash, with the performance differences being on the order of 2%. I do appreciate the inclusion of these variants in the results section; I am just not sure that they support the claim that TF generation _“lead[s] to less improvement.”_

2. On that note, are the authors surprised that the gap between AR and TF is so small, given the qualitative results in Figure 2 and distance metrics in Table 4?
3. How does Figure 2 differ if, instead of plotting all of the original data in blue, you only plot the last $T$ trajectories? Since these occur at the end of trajectories, you might expect these to be disjoint from the rest of the original data even if they are not resampled by a model.

Copy-editing:
L334: _"How much new trajectories"_ $\rightarrow$ "How many new trajectories"



**Limitations:**

As discussed in the “Strengths and Weaknesses” section, the main weakness is that BooT is a somewhat straightforward extension of existing work in model-based RL. However, the paper does a thorough investigation of this extension and does not overclaim its contribution.

**Strengths And Weaknesses:**

This paper considers a relatively straightforward combination of existing algorithms, but does so thoroughly. It is careful in its empirical evaluations, with baseline variants including (a) TT with extra training to isolate the effects of BooT’s extra gradient steps and (b) an S4RL-modified version of TT to compare the model-generated data to other forms of data augmentation. The paper is also careful not to overclaim its novelty, and combined with its thoroughness, certainly executes on the premise it describes in the abstract and introduction.

The section on “Further Analysis of Bootstrapping” hints at a question I have myself wondered about: when a model is trained on the same underlying data as a policy but can then generate more data that further improves the policy, what about the model-generated data caused that improvement? The answer is somewhat heuristic and qualitative here, but this question has underrecognized subtlety, so I appreciated an attempt at addressing it empirically here. I do think that the introduction might have overpromised a bit here, saying that the _“generated pseudo trajectories… [keep] consistent with the underlying MDP”_. Most would interpret that as meaning “consistent with the initial state distribution and transition function”, but the investigation the authors conducted doesn’t actually take into account dynamical feasibility but inter-state distances. That being said, this is mostly a quibble about terminology, and any attempt at answering this question at all is a good start.

---

> ### Author Response · Authors · 2022-08-02
> **Reply to Reviewer i5fj (Part 2)**
>
> ### Reviewer i5fj: Questions
>
> > "However, since teacher-forcing generation is not able to expand dataset coverage as effectively as autoregressive generation, it will lead to less improvement in performance."
> >
> > This explanation is what I would have expected, but does it actually pan out in the results? Table 2 shows teacher forcing (TF) underperforming autoregressive (AR) generation with “-once” sampling, but (TF) outperforming (AR) generation with “-repeat” sampling. Overall, though, it’s all a bit of a wash, with the performance differences being on the order of 2%. I do appreciate the inclusion of these variants in the results section; I am just not sure that they support the claim that TF generation “lead[s] to less improvement.”
>
> We agree that the difference between TF and AR results is not very sufficient to prove the claim that TF leads to less improvement than AR, indeed. So we have made our statement clear in the revised version. Besides, we also illustrated in Table 4 of Sec. 5.3 why AR is more effective than TF from the perspective of data distribution.
>
> The results in Table 4 show that, the distance from the generated data to the original offline data in AR is larger than TF, indicating AR is expanding the original offline data more aggressively (which might be due to the compounding error of the auto-regressive mechanism). Meanwhile, in the evaluation phase, the distance from the generated trajectories produced by AR to the real trajectories from the real environment is *smaller* than TF. This means AR is more capable of generating data consistent with the real distribution in RL tasks.
>
> After all, AR and TF are both our proposed new methods, achieving better results than baselines, and both prove the effectiveness of our algorithm.
>
> ---
>
> > On that note, are the authors surprised that the gap between AR and TF is so small, given the qualitative results in Figure 2 and distance metrics in Table 4?
>
> We are a bit surprised, but this can also be explained to some extent. As Figure 2 demonstrates, AR tends to expand the data coverage, while TF tends to complement the sparse part in the original dataset, where both methods somehow deviate from the original data distribution in Figure 2, and both are effective in improving the performance. Additionally, the amount of generated data used to train the model is not quite large, ranging from approximately 2% to 10% of the original data. As a result, it is somewhat reasonable that the gap in the improvement of these two methods is not large.
>
> ---
>
> > How does Figure 2 differ if, instead of plotting all of the original data in blue, you only plot the last trajectories? Since these occur at the end of trajectories, you might expect these to be disjoint from the rest of the original data even if they are not resampled by a model.
>
> Thanks for your good suggestions. We plot the complete trajectory, rather than only the generated part, in the figure: https://user-images.githubusercontent.com/110915925/183668952-77b608e7-8d10-4193-bbef-4d58b82d15ef.png
>
> The figure shows that the distribution of data from TF and AR differs from the distribution of original data, demonstrating that the generated data is effective in expanding the dataset. Meanwhile, the distribution of TF and AR gets similar, but still with some minor differences (as marked inside the box in the figure below). The reason for this might be that the first $(T-T')$ data are exactly the same as the data from TF and AR, thus making their distribution similar to each other.
>
> ---
>
> > Copy-editing: L334: "How much new trajectories"  "How many new trajectories"
>
> Thanks for pointing out this typo. We have fixed that in our revised version.
>
> ### Reviewer i5fj: Limitations
>
> > As discussed in the “Strengths and Weaknesses” section, the main weakness is that BooT is a somewhat straightforward extension of existing work in model-based RL. However, the paper does a thorough investigation of this extension and does not overclaim its contribution.
>
> Thank you for your reviews, and we are highly grateful for your recognition of our paper!

---

> ### Author Response · Authors · 2022-08-02
> **Reply to Reviewer i5fj (Part 1)**
>
> We are greatly thankful to the reviewer for the comments. We hope the following replies resolve the concerns raised in the review.
>
> ### Reviewer i5fj: Comments
>
> > The section on “Further Analysis of Bootstrapping” hints at a question I have myself wondered about: when a model is trained on the same underlying data as a policy but can then generate more data that further improves the policy, what about the model-generated data caused that improvement? The answer is somewhat heuristic and qualitative here, but this question has underrecognized subtlety, so I appreciated an attempt at addressing it empirically here. I do think that the introduction might have overpromised a bit here, saying that the *“generated pseudo trajectories… [keep] consistent with the underlying MDP”*. Most would interpret that as meaning “consistent with the initial state distribution and transition function”, but the investigation the authors conducted doesn’t actually take into account dynamical feasibility but inter-state distances. That being said, this is mostly a quibble about terminology, and any attempt at answering this question at all is a good start.
>
> We appreciate very much for your valuable comments. The sequence model used in TT and BooT is capable of generating a complete trajectory once given the initial state. It does not explicitly distinguish the "model" and the "policy" part and contains the transition dynamics as part of this model. Actually, it can simultaneously generate the next state like a "model" given past trajectories, and generate actions as a "policy".
>
> For the question about the consistency of the MDP, we think it is a reasonable choice to investigate the inter-state distances from both the dataset and the environment, as in our experiments in Sec. 5.3. In the evaluation phase (i.e., Environment in Table 4), the inter-state distance is a reasonable choice to estimate the difference between the learned transition dynamics and the real transition probability, as it is essentially calculating
>
> $\text{Distance} = \left\| \hat{s}\_{t+1} - s\_{t+1} \right\|^2, \quad \hat{s}\_{t+1} \sim \tilde{P}(s'|s=s\_t,a=a\_t), \quad s\_{t+1} \sim P(s'|s=s\_t,a=a\_t),$
>
> where $\tilde{P}(s'|s,a)$ is the learned model and $P(s'|s,a)$ is the ground-truth transition probability. This gives an estimation of the error between the learned transition dynamics and the real ones.
>
> Moreover, BooT will not generate the initial states (i.e., generated length $T'$ will be less than the orginal sequence length $T$). Thus, we can say that the initial state distribution of the generated trajectories is aligned with the original ground truth.
>
> In our opinion, the results in Table 4 can prove that our model is capable of somehow learning the underlying MDP: it predicts the next state relatively more accurately, as the results in Table 4 show, and it can also predict the actions as a "policy". Overall speaking, we think our analysis of the distribution of trajectories can partly demonstrate the model's capability of generating data consistent with the underlying MDP. It is also possible to manually create a simple environment, and compare the real underlying MDP with the learned probability. We will take this as an important future work to further explore the rationality of our proposed method. We thank again for your valuable suggestions.

---

### Official Review · Reviewer_9mfT · 2022-07-11

**Rating:** 5
**Confidence:** 3
**Soundness:** 2 fair
**Presentation:** 3 good
**Contribution:** 2 fair

**Summary:**

In order to tackle the problem that offline RL data is often small in size and lacks in coverage, this paper proposes to treat RL as a sequence modeling task and augment the training data by selecting the high confidence generated samples from the sequence model that is going through training.

The proposed method is evaluated on a subset of the D4RL continuous control tasks and is shown to outperform several baselines (including transformer-based and model-based ones as well as CQL and BC). for a bit over half of the tasks.


**Questions:**

* The method seems to be more stable than TT judging from table 1. Any idea why?
* I wonder if the authors have done experiments where they use this only as an extra source of data and train CQL (or other offline-RL methods) with the extra data?
* For teacher-forcing generation (l180-183), Is that only for one token, or one timestep, or something else? Either way it sounds quite limiting and the sampling method is yet another hparam that needs to be evaluated.
* Did you perform experiments on other datasets in D4RL gym? E.g. the “random” and “expert” datasets.

**Limitations:**

There are quite a few additional limitations not mentioned in the paper (l349-350): 1. The extra two hyperparameters introduced k and η require finetuning, which depends on availability to the environment or a good OPE method. 2. As mentioned in l37-39, for other tasks in general, it is unclear whether the dataset available is sufficient to train a BooT, unless we try it, which will incur extra training time and cost, as mentioned in l349-350.

**Strengths And Weaknesses:**


Strengths:
* The algorithm's main body is simple and orthogonal to other offline-RL works. The only constraint is to use a sequence model.
* Interesting analysis of the data distribution generated by teacher-forcing vs auto-regressive.
* Clear description of the method.

Weakness:
* It is unclear whether the gain of BooT comes from 1. Extra data 2. Different architecture (pretrained gpt2 vs not) 3. Some inherent property in the sequence model as opposed to other world models that may only predict the observation and the reward.
* It is unclear from the paper whether bootstrapping is novel beyond supervised learning (e.g. in RL)

---

> ### Author Response · Authors · 2022-08-02
> **Reply to Reviewer 9mfT (Part 3)**
>
> ### Reviewer 9mfT: Limitations
>
> > There are quite a few additional limitations not mentioned in the paper (l349-350): 1. The extra two hyperparameters introduced k and η require finetuning, which depends on availability to the environment or a good OPE method. 2. As mentioned in l37-39, for other tasks in general, it is unclear whether the dataset available is sufficient to train a BooT, unless we try it, which will incur extra training time and cost, as mentioned in l349-350.
>
> We appreciate your detailed review very much and pointing out some limitations of our work.
>
> 1. We fully agree that careful parameter tuning requires a real environment or OPE method for most offline RL methods. To alleviate this problem, we show that our method is not very sensitive to the additional hyperparameters. As the ablation study in Sec. 5.4 shows, most choices of $k$ and $\eta$ will lead to improvement in the performance compared to TT baseline, suggesting that our method is quite robust to these hyperparameters.
>
> 2. With decent experiment results on many most commonly used benchmark datasets, we have already shown that BooT can be helpful for various kinds of datasets, which have different properties (medium-expert, medium, medium-replay), different amounts of data (200k, 1m, 2m timesteps) and different tasks (HalfCheetah, Hopper, Walker2D). We believe that these results are sufficient to show the effectiveness of our method, as the same as most of the current offline RL works, which also evaluate their methods on these benchmarks. We also recognize that the training time cost is a limitation of our work, and we plan to improve it by incorporating more advanced techniques, such as non-autoregressive sequence generation, as we have discussed in the last section of our paper.
>
> ### Reviewer 9mfT: Referrence
>
> [1] Janner Michael, Li Qiyang, and Levine Sergey. Offline reinforcement learning as one big sequence modeling problem. In NeurIPS, 2021.
>
> [2] Chen, Lili, et al. "Decision transformer: Reinforcement learning via sequence modeling." Advances in neural information processing systems 34 (2021): 15084-15097.
>
> [3] Qizhe Xie, Minh-Thang Luong, Eduard Hovy, and Quoc V Le. Self-training with noisy student improves imagenet classification. In CVPR, 2020.

---

> > ### Comment · Reviewer_9mfT · 2022-08-07
> > **further comments to authors' reply**
> >
> > Thank you authors for the comments and the additional experiments. I’d like to follow up on several of those comments
> >
> > > It is unclear whether the gain of BooT comes from 1. Extra data 2. Different architecture (pretrained gpt2 vs not) 3. Some inherent property in the sequence model as opposed to other world models that may only predict the observation and the reward.
> >
> > Thank you authors for the clarification. I think it is clear to me now that the better performance comes from 1. Training on boostrapped data with some generation percentage η%, 2. The exact way of generating such data, which is either teacher forcing or auto-regressive.
> >
> > > For teacher-forcing generation (l180-183), Is that only for one token, or one timestep, or something else? Either way it sounds quite limiting and the sampling method is yet another hparam that needs to be evaluated.
> >
> > I understand that you concatenate the predictions together where the input is always the original sequence – the typical setup for teacher-forcing. That results in a generated trajectory τ˜>T −T′ where each timestep and each token is generated independently from one another. As a result, you will have a generated sequence that completely ignores the actions taken, because the reward r and the next state s’ are generated based on the ground truth action, not the predicted action. And by training on such data, somehow the transformer is still fine. That part does not make sense to me.
> >
> > My comment around the hyperparameter is more about the choice between TF and AR. e.g. you proposed four closely related methods: BooT-o, AR; BooT-r, AR; BooT-o, TF; BooT-r, TF. That is 4x the number of offline policy evaluations to do if you plan to use this in the real world. If AR performs better or similarly to TF most of the time (which seems like the case from table 2 and 4), then you can suggest the reader to do AR and save the effort of trying out TF, which costs extra training time and evaluation effort.
> >
> > > Did you perform experiments on other datasets in D4RL gym? E.g. the “random” and “expert” datasets.
> >
> > Thank you for providing the numbers. It is included in quite a lot of model-based offline RL works like MOPO [1] (which does quite well on random) and MBOP[2]. So please consider at least including that in appendix.
> >
> > [1] Yu, Tianhe, et al. "Mopo: Model-based offline policy optimization." Advances in Neural Information Processing Systems 33 (2020): 14129-14142.
> > [2] Argenson, Arthur, and Gabriel Dulac-Arnold. "Model-based offline planning." arXiv preprint arXiv:2008.05556 (2020).

---

> > > ### Author Response · Authors · 2022-08-08
> > > **Further Reply to Reviewer 9mfT**
> > >
> > > ## Reviewer 9mfT: Further reply
> > >
> > > > I understand that you concatenate the predictions together where the input is always the original sequence – the typical setup for teacher-forcing. That results in a generated trajectory τ˜>T −T′ where each timestep and each token is generated independently from one another. As a result, you will have a generated sequence that completely ignores the actions taken, because the reward r and the next state s’ are generated based on the ground truth action, not the predicted action. And by training on such data, somehow the transformer is still fine. That part does not make sense to me.
> > >
> > > We fully agree with you that, the generated sequence by TF (teacher-forcing) mechanism ignores the actions taken here. However, we would like to re-emphasize that BooT is based on the seuqence modeling framework for offline RL, which is quiet different from some conventional model-based RL methods that explicitly model the transition probability or reward function. Thus, BooT-TF is more similar to applying perturbations on the training data by generation, and further augment training.
> > >
> > > Compared to the previous perturbation-based method S4RL which simply adds random noise on the input states, our method applys perturbations on the whole trajectory data, including states, actions and rewards. We believe that, the perterbation on the whole original trajectory data could expand the coverage of dataset more effectively, as shown in Table 1 in our paper. Besides, using generating rather than random noise to perturb also makes the perturbed data not deviating from the underlying MDP very much, as demonstrated in Table 4 in our paper.
> > >
> > > As a result, TF generation of our BooT method still achieves improvement compared to the baselines.
> > >
> > > ---
> > >
> > > > My comment around the hyperparameter is more about the choice between TF and AR. e.g. you proposed four closely related methods: BooT-o, AR; BooT-r, AR; BooT-o, TF; BooT-r, TF. That is 4x the number of offline policy evaluations to do if you plan to use this in the real world. If AR performs better or similarly to TF most of the time (which seems like the case from table 2 and 4), then you can suggest the reader to do AR and save the effort of trying out TF, which costs extra training time and evaluation effort.
> > >
> > > We agree that it is quite reasonable to focus on one particular scheme if it performs better in most cases. Our suggestion for the choice between TF and AR is that, people would better choose TF schema if the generation length or the number of tokens per timestep is large (to save inference costs), otherwise people should choose AR schema for better performance, as AR generation consumes a large amount of time while might achieve relatively better performance. These differences between TF and AR have been included in our paper (L184-L190). As for the usage of the generated data, we found that "Bootstrap-once" or "Bootstrap-repeat" have little performance difference. Thus, we do not recommend preference for this setting.
> > >
> > > The reason why we proposed four different schemes in BooT is to help conducting a comprehensive study on our proposed method, i.e., bootstrapped transformer for offline reinforcement learning. We expect this would provide a thorough exploration and some guidelines to help the reader to conduct practical usage of our method.
> > >
> > > ---
> > >
> > > > Thank you for providing the numbers. It is included in quite a lot of model-based offline RL works like MOPO (which does quite well on random) and MBOP. So please consider at least including that in appendix.
> > >
> > > Thanks for your comment! We have added the results and discussions on 'random' and 'expert' datasets in the revised paper appendix.
> > >
> > > ---
> > >
> > > We sincerely look forward to your reply to our response, and we are open to any discussion to improve our paper.
> > >
> > > Best wishes!
> > >
> > > The authors.

---

> > > > ### Comment · Reviewer_9mfT · 2022-08-09
> > > > **Response and updating scores**
> > > >
> > > > > Compared to the previous perturbation-based method S4RL which simply adds random noise on the input states, our method applys perturbations on the whole trajectory data, including states, actions and rewards.
> > > >
> > > > This perspective is interesting. Good to see that it is better than random perturbation, but currently the main knobs to control this perturbation are the generation percentage in Algorithm 1 and the sampling method. I could foresee future improvements where you do finer level control of that.
> > > >
> > > >  > The reason why we proposed four different schemes in BooT is to help conducting a comprehensive study on our proposed method, i.e., bootstrapped transformer for offline reinforcement learning. We expect this would provide a thorough exploration and some guidelines to help the reader to conduct practical usage of our method.
> > > >
> > > > Makes sense.
> > > >
> > > > ## Score update
> > > >
> > > > I decide to update the scores to reflect the new information gained during rebuttal and to encourage further research along this line of work. Thank you authors for your detailed reply.

---

> ### Author Response · Authors · 2022-08-02
> **Reply to Reviewer 9mfT (Part 2)**
>
> ### Reviewer 9mfT: Questions
>
> > I wonder if the authors have done experiments where they use this only as an extra source of data and train CQL (or other offline-RL methods) with the extra data?
>
> Thanks for your constructive suggestion. We try to train CQL with additional data generated by BooT, with both TF and AR generation. The results are listed below.
>
> | Dataset | Game | BooT+CQL | BooT |
> |:---:|:--:|:--:|:--:|
> | Medium-expert | HalfCheetah | 5.0 | 94.0  |
> | Medium-expert | Hopper | 0.8 | 102.3 |
> | Medium-expert | Walker2d | 26.4 | 110.4 |
> | Medium | HalfCheetah | 30.0 | 50.6 |
> | Medium | Hopper | 79.8 | 70.2 |
> | Medium | Walker2d | 6.4 | 82.9 |
> | Medium-replay | HalfCheetah | 4.3 | 46.5 |
> | Medium-replay | Hopper | 5.0 | 92.9 |
> | Medium-replay | Walker2d | 5.8 | 87.6 |
> |  **Average**  | | 18.2 | 81.9 |
>
> From the results, we can see that CQL does not perform with generated data from BooT. We think the reason mainly lies in two aspects:
>
> 1. It is non-trivial to utilize the generated data from BooT to train CQL, as CQL models the continuous data distribution, and using generated data requires the inverse operation of tokenization, which causes information loss and needs to be carefully treated. However, our BooT model relies on Transformer architecture and models the distribution of discretized data and does not require such inverse operation.
>
> 2. BooT is a self-improving method, using the generated data to improve the sequence model itself further. However, using the generated data from BooT to train other models makes BooT more like a generative model and introduces more hyperparameters, such as the ratio of the generated data in CQL training, making the model much harder to train. This is beyond the scope of our work. However, we believe it is an interesting idea, and we will leave it as future work.
>
> ---
>
> > For teacher-forcing generation (l180-183), Is that only for one token, or one timestep, or something else? Either way it sounds quite limiting and the sampling method is yet another hparam that needs to be evaluated.
>
> We guess there is some misunderstanding to our method. We generate the last $T'$ timesteps for each input trajectory, as explained in Eq. (5) and lines 167-171. Each timestep corresponds to one transition, including ($s_t, a_t, r_t, R_{t+1}$) as shown in Figure 1, which contains $(N+M+2)$ tokens as explained in Line 139-143. Moreover, each timestep in a complete trajectory in the original dataset, except for the first timestep, could be generated in our training procedure. Besides, the sampling procedure in our generation process uses the same hyperparameters as TT [1] in its evaluation phase. As a result, the trajectory generation part only requires one additional hyperparameter $T'$ and can be applied to any timestep in a trajectory without additional limitation.
>
> ---
>
> > Did you perform experiments on other datasets in D4RL gym? E.g. the “random” and “expert” datasets.
>
> We performed experiments on both the random and the expert dataset, and the results are listed below. We listed the results of the baselines from their corresponding original paper if the baseline runs experiments on these two datasets.
>
> | Dataset | Game | TT | BooT-o, AR | BooT-r, AR | BooT-o, TF | BooT-r, TF |
> |:-- |:-- | --:| --:| --:| --:| --:|
> | Expert | HalfCheetah |  95.3 | 92.3 | 94.4 | 95.0 | **95.4** |
> | Expert | Hopper | 102.3 | 110.3 | **110.5** | 104.6 | 108.2 |
> | Expert | Walker2D | 108.4 | 108.5 | **108.7** | 108.5 | 108.5 |
> | Average | | 102.0 | 103.7 | **104.6** | 102.7 | 104.1 |
> | Random | HalfCheetah | 7.9 | 6.7 | 6.9 | 4.6 | 7.5 |
> | Random | Hopper | 6.7 | 6.8 | 6.5 | 6.5 | 6.6 |
> | Random | Walker2D | 5.6 | 5.2 | 4.8 | 4.6 | 4.8 |
> | Average | | 6.8 | 6.3 | 6.1 | 5.2 | 6.3 |
>
> We cannot see significant differences between the scores of methods on these datasets. This is as expected since expert datasets are usually used to test imitation learning algorithms, but not offline RL algorithms; and the quality of random datasets is too poor, and thus little valuable information can be extracted from them. These are also the reasons that pure expert and random datasets are usually not included in many previous offline RL works [1, 2], and we follow their experimental settings.

---

> ### Author Response · Authors · 2022-08-02
> **Reply to Reviewer 9mfT (Part 1)**
>
> We are greatly thankful to the reviewer for the comments. We hope the following replies resolve the concerns raised in the review.
>
> ### Reviewer 9mfT: Comments
>
> > It is unclear whether the gain of BooT comes from 1. Extra data 2. Different architecture (pretrained gpt2 vs not) 3. Some inherent property in the sequence model as opposed to other world models that may only predict the observation and the reward.
>
> Thanks for your valuable questions. We would like to make us more clear as following:
>
> 1. (a) We use the same dataset and guarantee fair comparison among all the tested baselines. For the generated training data, we would prefer to regard them as a part of our BooT framework, since they are generated by the model itself without requiring other information. This is exactly evidence of the effectiveness of our algorithm.
>
>    (b) We guess you might mean the number of gradient steps in the training procedure. Compared to Trajectory Transformer [1] (TT) trained with the same number of extra training steps, named TT (Re-train) in Table 1 (69.4), our method BooT obtains much better results (81.9). This shows that the improvement is not from extra training gradient steps.
>
> 2. We use the same model architecture as TT, *without* using pretrained models or any other training resources. Compared to TT, we obtain $16.8\%$ improvement with exactly the same model architecture. Moreover, compared to a strong baseline DT-gpt2, which adopts Decision Transformer [2] (DT) algorithm using the pretrained GPT-2 model, our method BooT has achieved better results with an 80 times smaller model architecture than that. All these observations could sufficiently prove that the improvement of our proposed method BooT is not from the difference between architectures.
>
> 3. As we have mentioned, BooT achieves an improvement of $16.8\%$ compared to TT. Considering BooT and TT utilize the same model architecture, this could exclude the impact of different model properties between the sequence model and other world models.
>
> In conclusion, we have illustrated that the improvement of BooT actually comes from our proposed algorithm, which contains trajectory data generation and utilization frameworks rather than some other factors.
>
> ---
>
> > It is unclear from the paper whether bootstrapping is novel beyond supervised learning (e.g. in RL)
>
> To the best of our knowledge, we are the first to try bootstrapping from fully generated data in RL. We would like to point out that, though the word "bootstrapping" is also used in Q-learning, referring to calculating the Bellman target with a sampled next action, it is totally different from our proposed method. Furthermore, we are generating sequence trajectory data, rather than pseudo-labels as utilized in supervised learning tasks such as [3]. Moreover, the difficulty and complexity of bootstrapping on RL are beyond that in supervised learning. That is why we think BooT is novel beyond other bootstrapping methods in supervised learning.
>
> ### Reviewer 9mfT: Questions
>
> > The method seems to be more stable than TT judging from table 1. Any idea why?
>
> Thanks for pointing this out! This might be another advantage of BooT that we haven’t noticed yet. From Table 1, we can see that deviations of 6 (of 9) tasks are reduced with more gradient steps (TT Re-train), and the learning can be further stabilized with more gradient steps produced by generated data. The potential reason might be that the quality of generated data is higher. Specifically, generated data can remedy the limitation of the original data distribution, as illustrated in Figure 2 in our paper. Moreover, as is shown in Table 4 in the paper, the generated data are closer to the ground truth trajectories from the real environment in the evaluation, which illustrates that the generation somehow follows the underlying MDP. These properties make room for improving the robustness of TT. Our method BooT helps remedy the problem of insufficient training data and eventually improves the robustness of the derived transformer model.

---

### Official Review · Reviewer_hqzJ · 2022-07-12

**Rating:** 6
**Confidence:** 4
**Soundness:** 3 good
**Presentation:** 4 excellent
**Contribution:** 3 good

**Summary:**

This paper proposes Bootstrapped Transformer, which uses bootstrapping to self-generate more offline data to improve sequence model training of Trajectory Transformer in offline RL setting. The authors study autogenerative generation and teacher forcing for trajectory generation in bootstrapping along with data filtering based on confidence score (prediction probability) and simple curriculum learning. They demonstrate better performance comparing to other offline RL baselines in D4RL dataset, specifically on Gym-Mujoco and Adroit. They also conduct analysis on bootstrapped data from the perspective of coverage/diversity and alignment with MDP, and an ablation to provide simple guideline of bootstrapping.

**Questions:**

Please check weakness in the previous section. Specifically, my major concerns lie in
- Some straightforward potential improvement/alternative of baselines.
- How does the proposed method perform under non-expert demonstration given bootstrapping greatly relies on the "correctness" of the original model?
- Further discussion and empirical evidence for autoregressive generation.

**Limitations:**

Apart from large training time due to online pseudo training data generation mentioned by the authors, it would be nice to see more details or even failure mode resulting from instability of bootstrapping. The authors mentioned some of them in ablation as well as techniques to remedy such issue, but if any more, it would be very helpful to see all other failure strategies that the authors have tried.

**Strengths And Weaknesses:**

Strengths
- The paper is well-written and easy to follow.
- The proposed method is well-motivated and the problem being attacked is important in using sequence modeling in offline RL.
- The main results are in general quite good and support the major contribution of this work.

Weakness
- TT-reproduce is somewhat very different from TT-original; actually, it's worse in general. Thus I wonder if it's possible that TT-original with retraining can achieve better results.
- Why S4RL used in the experiment only considers Gaussian noise but not also other augmentation techniques proposed in the original paper like dropout, mixup, etc.?
- Regarding TT-retrain baseline, additional to uniformly sample from the original dataset for additional training, what about a simple weighted sampling based on loss or data likelihood?
- The author claims in line 184-190 that autoregressive generation is better than teacher forcing due to its more diverse generation. In spite of the seemingly persuasive explanation, the marginal difference in Table 2 plus no BooT AR in Adroit experiment (Table 3; though the author mentions it is too time-consuming, I wonder if there is an alternative to provide more concrete justification) don't well justify the statement.
- From Figure 2, AR and TF seem to be complementary to each other (with TF covers sparser regions in the original data distribution). I wonder combining these two bootstrapping techniques (with simple methods like randomly choosing one approach to generate data) will provide further improvement.
- How does the proposed method perform under non-expert demonstration? Specifically, bootstrapping largely depends on the "correctness" of the original model (i.e., TT here). I wonder if we gradually mix less-performing data to the original training set (e.g., replacing some portion of dataset with "random" dataset in D4RL), how will this negatively affect bootstrapping?
- While autoregressive generation can produce larger variation, how does the accumulated error affect bootstrapping? Autoregressive generation with larger generation length (T') produces data with larger deviation from the original dataset but at the same time suffers from larger accumulated error. Thus, T' may be an interesting hyperparameter for ablation study to see how to strike a good balance between the tradeoff.

---

> ### Author Response · Authors · 2022-08-02
> **Reply to Reviewer hqzJ (Part 3)**
>
> ### Reviewer hqzJ: Comments
>
> > While autoregressive generation can produce larger variation, how does the accumulated error affect bootstrapping? Autoregressive generation with larger generation length ($T'$) produces data with larger deviation from the original dataset but at the same time suffers from larger accumulated error. Thus, $T'$ may be an interesting hyperparameter for ablation study to see how to strike a good balance between the tradeoff.
>
> We would also like to see the impact of generation length ($T'$) on the performance. However, as the experiments of AR generation takes much time, we are only able to provide TF ablation results at this time for reference. We will try our best to complete the AR ablation experiments with a smaller range of $T'$ and provide them in a separate response. As $T'$ increases, the overall generated trajectories will tend to deviate more from the original sequence even with TF generation. Thus, we expect the results have, to some extent, addressed your concern.
>
> The TF results are shown in the figure in the link: https://user-images.githubusercontent.com/110915925/183668736-71650288-2811-4375-9986-205b02ae8063.png
>
> Recall that the generated part length $T' < T = 10$, where $T$ is the length of the original trajectory. From the figure, we can see that the performance of BooT with TF generation generally decreases as $T'$ increases, and most results are better than the baselines.
>
> Furthermore, the performance of BooT-r is relatively robust to the changes in $T'$. As the number of tokens per timestep differs between different environments, it is difficult to determine the most suitable $T'$ for each environment. As a result, we think it reasonable to simply set $T'=1$ in our experiments.
>
> ### Reviewer hqzJ: Question
>
> **Q1:** Some straightforward potential improvement/alternative of baselines.
>
> **A1:** We have provided some extra baseline experiments on the above, and explained the reason why we did not choose some other baseline settings.
>
> ---
>
> **Q2:** How does the proposed method perform under non-expert demonstration given bootstrapping greatly relies on the "correctness" of the original model?
>
> **A2:** We have provided additional experiments and our analysis above.
>
> ---
>
> **Q3:** Further discussion and empirical evidence for autoregressive generation.
>
> **A3:** We have provided the experiments and our analysis above.
>
> ### Reviewer hqzJ: Limitations
>
> > Limitations: Apart from large training time due to online pseudo training data generation mentioned by the authors, it would be nice to see more details or even failure mode resulting from instability of bootstrapping. The authors mentioned some of them in ablation as well as techniques to remedy such issue, but if any more, it would be very helpful to see all other failure strategies that the authors have tried.
>
> One of the failure strategies we have tried is the confidence filtering technique. In our early experiments, we also tested different filtering techniques other than the current one. At first, we did not apply any filtering techniques to generated data, and obtained a mean result of 72.7. We then tried applying a hard confidence threshold, i.e., only choose generated data with confidnce above a hard threshold, where the threshold is a hyperparameter to be tuned. This method obtained a mean results of 71.5, and we discovered that there are often cases where too many or too few generated data that is filtered (almost all generated data are above the threshold, or no data are above the threshold). Both two techniques result in similar std. in results compared to other TT baselines and higher std. compared to TT. We finally decided to use generated data with top-$\eta\%$ confidence. This filtering technique provides the best results, and the hyperparameter is also relatively easier to finetune compared to the hard confidence threshold.
>
> ### Reviewer hqzJ: Reference
>
> [1] Janner Michael, Li Qiyang, and Levine Sergey. Offline reinforcement learning as one big sequence modeling problem. In NeurIPS, 2021.
>
> [2] Sinha, Samarth, Ajay Mandlekar, and Animesh Garg. "S4RL: Surprisingly simple self-supervision for offline reinforcement learning in robotics." Conference on Robot Learning. PMLR, 2022.

---

> > ### Comment · Reviewer_hqzJ · 2022-08-09
> > **Response**
> >
> > Thanks to the authors for detailed discussion and extensive additional experiments. The rebuttal addressed most of my previous concerns. I would like to raise my final rating.

---

> ### Author Response · Authors · 2022-08-02
> **Reply to Reviewer hqzJ (Part 2)**
>
> ### Reviewer hqzJ: Comments
>
> > The author claims in line 184-190 that autoregressive generation is better than teacher forcing due to its more diverse generation. In spite of the seemingly persuasive explanation, the marginal difference in Table 2 plus no BooT AR in Adroit experiment (Table 3; though the author mentions it is too time-consuming, I wonder if there is an alternative to provide more concrete justification) don't well justify the statement.
>
> We apologize for the confusion. In addition to our explanation in lines 184-190, we also illustrated in Table 4 of Sec. 5.3 why AR is more effective than TF from the perspective of data distribution.
>
> The results in Table 4 show that, the distance from the generated data to the original offline data in AR is larger than TF, indicating AR is expanding the original offline data more aggressively (which might be due to the compounding error of the auto-regressive mechanism). Meanwhile, in the evaluation phase, the distance from the generated trajectories produced by AR to the real trajectories from the real environment is *smaller* than TF. This means AR is more capable of generating data consistent with the real distribution of the underlying MDP in RL tasks.
>
> After all, AR and TF are both our proposed new methods, achieving better results than baselines, and both prove the effectiveness of our algorithm. We have made our statement clear in the revised version.
>
> ---
>
> > From Figure 2, AR and TF seem to be complementary to each other (with TF covers sparser regions in the original data distribution). I wonder combining these two bootstrapping techniques (with simple methods like randomly choosing one approach to generate data) will provide further improvement.
>
> Thanks for your excellent suggestion! We agree with you that using the generated data with both TF and AR generation might be helpful in improving performance. However, we would like to put more attention to demonstrating our BooT framework, whose effectiveness is proved by either TF or AR generation. Combining TF and AR generation would certainly, in intuition, benefit the performance, but it also introduces additional hyperparameters, such as the combining proportion of the data from the two mechanisms, thus increasing the difficulty of hyperparameter tuning. This is why we did not conduct such experiments in our main paper.
>
> For clarity, we will also perform experiments using both TF and AR generation. Due to limited time and resources, we expect to provide the results in the next few days in a separate response, and we are afraid that we cannot perform a complete hyperparameter tuning, and the results might still have the potential to be improved.
>
> ---
>
> > How does the proposed method perform under non-expert demonstration? Specifically, bootstrapping largely depends on the "correctness" of the original model (i.e., TT here). I wonder if we gradually mix less-performing data to the original training set (e.g., replacing some portion of dataset with "random" dataset in D4RL), how will this negatively affect bootstrapping?
>
> We think it is also a good idea to test the impact of less-performing data on our algorithm. We try to replace $\phi$% of the medium-replay dataset with the random dataset and perform experiments on HalfCheetah, Hopper, and Walker2d environments. Our reproduced results are listed below. The results demonstrate that, in general, the performance of all methods decreases as the ratio of random data increases. Among all methods, *BooT-r, AR* achieves the best results until using a pure random dataset, when the performance of all methods drops to the same level.
>
> | $\phi$% | TT | BooT-o, AR | BooT-r, AR | BooT-o, TF | BooT-r, TF |
> | -- | -- | -- | -- | -- | -- |
> | $20$% | 57.2 | 50.8 | **66.0** | 51.0 | 55.9 |
> | $40$% | 53.7 | 47.6 | **61.6** | 39.6 | 39.0 |
> | $60$% | 32.5 | 25.7 | **44.8** | 17.6 | 28.0 |
> | $80$% | 11.1 | 10.2 | 12.8 | **13.8** | 11.5 |
> | Pure Random | **6.8** | 6.3 | 6.1 | 5.2 | 6.3 |
>
> It is also worth noting that the "medium-replay" dataset in the D4RL benchmark contains all the trajectories in the replay buffer of training an agent from scratch to a medium level. This means that the first several trajectories in the medium-replay dataset are close to random trajectories and are less-performing, which has, to some extent, already satisfied the situation, as mentioned in your review comments. In the experiments listed above, by replacing part of the medium-replay dataset with the random dataset, we increase the ratio of less-performing data.

---

> > ### Author Response · Authors · 2022-08-08
> > **Supplementary Experiment Results for Reply (Part 2)**
> >
> > For the second comment in Reply (Part 2), we mentioned that we also performed experiments using both TF and AR generation, denoted as *TF+AR*. Due to limited time and resources, we are sorry that we cannot perform a complete hyperparameter tuning, and the results might still have the potential to be improved. In the experiments, we set the combining proportion to 1:1, i.e. the number of generated data with TF is equal to AR, and we set the generation threshold $k=0.4$, which is the best one in our ablation study. The results are listed as below:
> >
> > | TT (Original)  | TT (Reproduce) | BooT-o, TF+AR  | BooT-r, TF+AR  |
> > | -------------- | -------------- | -------------- | -------------- |
> > | 72.6           | 70.1           | 74.7           | 77.8           |
> > | **BooT-o, AR** | **BooT-r, AR** | **BooT-o, TF** | **BooT-r, TF** |
> > | 81.9           | 79.1           | 79.9           | 80.9           |
> >
> > We can see that, all these BooT settings of our method have achieved relatively better performance than the baseline. Though that, using TF and AR generation *simultaneously* (TF+AR) does not provide an improvement in performance, comparing to that using pure TF or AR generation scheme individually.
> >
> > We should mention that combining the two schemas together (TF+AR) is non-trivial which also incorporates additional hyperparameters such as mixing ratio of the data from the two generation schemas. In this experiment, we just simply mix the two data sources as 50% + 50% to validate the possibility of this mixture boostrapping idea. Hopefully, we think that it could obtain better results with *TF+AR* if we further explore and refine the mixing algorithm on *TF+AR* setting, yet it has exceeded the scope of our paper and we leave it as a future work. We consider individual TF or AR generation have provided significant improvement and illustrated the superiosity of our proposed method, i.e., bootstrapped transformer for offline reinforcement learning.

---

> ### Author Response · Authors · 2022-08-02
> **Reply to Reviewer hqzJ (Part 1)**
>
> We are greatly thankful to the reviewer for the comments. We hope the following replies resolve the concerns raised in the review.
>
> ### Reviewer hqzJ: Comments
>
> > TT-reproduce is somewhat very different from TT-original; actually, it's worse in general. Thus I wonder if it's possible that TT-original with retraining can achieve better results.
>
> The results of "TT-original" were just taken from the original paper of Trajectory Transformer (TT) [1]. We reproduce the results of "TT-reproduce" with the same open-source code of TT.
> And "TT-retrain" means that the training set of TT has been expanded by sampling more from the original offline dataset and keeping the whole training data size the same as that of BooT for a fair comparison.
> Note that, the "TT-reproduce" and "TT-retrain" experiments are both based on the source code of TT [1], without modifying any part of the codes except for re-training. Meanwhile, the gap between our reproduced result of TT (70.1) and the reported result from TT's original paper (72.6) is not very large.
> We guess the "TT-retrain" method in our paper (as shown in Table 1) might just be the TT-original with re-training technique, as mentioned in your comments.
> From our experiments in Table 1, simply retraining TT on the original offline datasets did not provide any improvements (TT-retrain: 69.4; TT-reproduce: 70.1), which further reflects the effectiveness of our bootstrapped transformer method (BooT: 81.9).
>
> ---
>
> > Why S4RL used in the experiment only considers Gaussian noise but not also other augmentation techniques proposed in the original paper like dropout, mixup, etc.?
>
> We only tried Gaussian noise in our experiments for mainly two reasons: its good performance in S4RL and its good feasibility to be applied on TT.
> - Firstly, adding Gaussian noise is the second best method in S4RL, as can be seen from Table 1 in [2], where the best method of adversarial state training is not applicable on TT as it requires the Q-value estimation. Moreover, compared to some other methods like amplitude scaling (51.60), Gaussian noise (60.57) achieves much better results.
> - Secondly, some method like dropout is partly conflicted with TT, as the Transformer model in TT already contains dropout layers, and we think it is unnecessary to explicitly dropout the input data. Also, some methods like state-switch will break the sequential relationship of the data. Considering the feasibility of different augmentation techniques in S4RL, we believe adding Gaussian noise is a good choice to conduct the comparison between our method and the offline data augmentation method.
>
> As a result, we only tried Gaussian noise, which has no conflict with TT and is the best-performing method applicable to TT.
>
> ---
>
> > Regarding TT-retrain baseline, additional to uniformly sample from the original dataset for additional training, what about a simple weighted sampling based on loss or data likelihood?
>
> Thanks for your valuable idea, and we think it is worth trying to perform TT-retrain experiments with weighted sampling. We try to add a simple loss-weighted re-train component for TT-retrain, choosing the data with the lowest / largest $\eta$% loss in each batch to re-train the model. As we are using the negative log-likelihood as our loss function, the loss-weighted re-train and likelihood-weighted re-train are essentially the same in our experiments. The results are listed below. We also provide a figure for better visualization of these experiment results.
>
> | $\eta$% | TT retrained with loss-weighted (largest) data | TT retrained with loss-weighted (lowest) data | Baselines | |
> |:-- | --:| --:|:-- | --:|
> | $2$% | 69.5 | 71.4 | TT (Original) | 72.6 |
> | $4$% | 66.9 | 71.8 | TT (Reproduce) | 70.1 |
> | $6$% | 66.8 | 67.4 | TT (Re-train) | 69.4 |
> | $8$% | 70.5 | 71.9 | BooT (ours) | **81.9** |
> | $10$% | 66.0 | 69.2 | | |
>
> Image: https://user-images.githubusercontent.com/110915925/183668879-c803de87-1974-4603-8eff-e07226c503b5.png
>
> The results have illustrated that using a loss-weighted re-train does help to improve the performance of the model, but BooT still performs much better than both schemes of loss-weighted re-train. Compared to simple random re-training scheme, re-training the model with data corresponding to the lowest loss helps improve the performance of the model, while using data corresponding to the largest loss has less improvement or even possibly degrades the performance.

---

### Author Response · Authors · 2022-08-08
**General Reply**

We greatly thank all the reviewers for the insightful comments and suggestions, which are very helpful for us to improve this work further. We are greatly encouraged by the positive comments of reviewers, e.g.,

- The paper is well-written and easy to follow (hqzJ)
- The paper has a clear description of the method (9mfT)
- The paper has considered a combination of existing algorithms thoroughly and is careful in its empirical evaluations (i5fj)

We have also added some extra explanations and experiments according to the reviewers' comments. We have further included all new experiments in a new section as Appendix C, and moved the content in original Appendix C to Appendix D. These major revisions are shown in blue fonts in the updated version of Appendix. The siginificant revisions are summarized as follows:

- We added extra explanation in Appendix D.1, explaining the details of inter-state distance calculation, according to the comment of reviewer i5fj.
- We added extra experiments on random and expert dataset, according to the comment of reviewer 9mfT.
- We added extra experiments on random-mixed dataset, replacing part of the medium-replay dataset with the random dataset, according to the comment of reviewer hqzJ.
- We added extra experiments on TT retrained with loss-weighted data, according to the comment of reviewer hqzJ.
- We added an additional ablation study on generation length $T'$ with TF generation, according to the comment of reviewer hqzJ.

All the additional experiments we provided correspond to 15 random seeds, as the same setting in our paper.

---

### Meta-Review · Area_Chair_vwi7 · 2022-08-29

**Recommendation:** Accept
**Confidence:** Certain

**Metareview:**

## Summary
Offline RL (RL) algorithms aim to learn policies without interacting with an environment purely from the state and actions covered in the offline datasets. However, in real-world datasets, the coverage can be insufficient to learn good policies. Thus it is an important research direction to improve the sample efficiency of those methods. This paper approaches offline RL from a sequence modeling perspective. The paper adopts a variant of trajectory transformers for data generation, and they investigate two of the main design decisions in those models:
* Sampling methods (autoregressive vs. teacher-forcing based).
* Reuse of model-generated data

The authors validate their idea on two D4RL tasks:
* adroit
* locomotion

## Decision

Overall the paper is well-written and easy to understand. The results and experiments are thorough, and the paper goes for more depth in the experiments rather than breadth. The ideas in the paper are not novel, but the paper does not overclaim its contributions, and results are interesting. As a result, I think both NeurIPS and the broader offline RL communities would benefit from the findings of this paper. I am nominating this paper for acceptance.

The reviewers were very positive about this paper during the rebuttal and discussion paper. They all agreed that the paper is valuable and interesting contribution to the community. The main criticism of this paper that came up during the discussion period was that the idea is just a straightforward combination of the existing techniques. However, the idea presented in the paper is coherent, reasonable, and executed well.

The authors provided a very detailed rebuttal with clarifications to the points that reviewers raised. As a result of the rebuttal, some of the reviewers increased their scores. I would recommend that the authors incorporate some of those clarifications into the camera-ready paper version. Some of those are:

* In response to reviewer 9mfT’s question on the results with CQL on additional data generated by Boot is very interesting. I think the authors should include it in the camera-ready version of the paper.
* Additional experiments on other datasets from the D4RL gym environment as asked by reviewer 9mfT.
* The experiments asked by the reviewer *hqzJ*.


**Award:**

No

---

### Decision · Program_Chairs · 2022-09-14

Accept